# Rigidity enhances a magic-number effect in polymer phase separation

Bin Xu [1], Guanhua He[2], Benjamin G. Weiner [1], Pierre Ronceray [3], Yigal Meir[4], Martin C. Jonikas [2] & Ned S. Wingreen [2,5 ✉]

Cells possess non-membrane-bound bodies, many of which are now understood as phase-separated condensates. One class of such condensates is composed of two polymer species, where each consists of repeated binding sites that interact in a one-to-one fashion with the binding sites of the other polymer. Biologically-motivated modeling revealed that phase separation is suppressed by a "magic-number effect" which occurs if the two polymers can form fully-bonded small oligomers by virtue of the number of binding sites in one polymer being an integer multiple of the number of binding sites of the other. Here we use lattice-model simulations and analytical calculations to show that this magic-number effect can be greatly enhanced if one of the polymer species has a rigid shape that allows for multiple distinct bonding conformations. Moreover, if one species is rigid, the effect is robust over a much greater range of relative concentrations of the two species.

[1] Department of Physics, Princeton University, Princeton, NJ 08540, USA. [2] Department of Molecular Biology, Princeton University, Princeton, NJ 08540, USA. [3] Princeton Center for Theoretical Science, Princeton University, Princeton, NJ 08540, USA. [4] Department of Physics, Ben Gurion University of the Negev, 84105 Beer Sheva, Israel. [5] Lewis-Sigler Institute for Integrative Genomics, Princeton University, Princeton, NJ 08540, USA. ✉email: wingreen@princeton.edu

In addition to membrane-bound organelles, cells possess non-membrane-bound bodies including nucleoli, P-bodies, and stress granules, which are now understood as phase-separated condensates[1–4]. Typically, the components of these condensates have a high rate of exchange with the surrounding medium and the condensates themselves are dynamic, rapidly assembling and disassembling in response to specific stimuli[5–7]. The relevant properties of components include the presence of intrinsically disordered regions, as well as the valence, strength, and specific sequence of interacting residues or domains.

One biologically relevant class of condensates are those composed of two species of multivalent polymers or particles with specific interactions that drive phase separation[8–10]. In the simplest case, each component consists of repeated domains that interact in a one-to-one fashion with the domains of the other component. Such two-component multivalent condensates have been observed in several natural and engineered contexts. One example, the algal pyrenoid, is a carbon-fixation organelle, in which the two components essential for assembly[11] are the rigid oligomer Rubisco (the active enzyme in $CO_2$ fixation) and EPYC1, an unstructured linker protein[12]. Another multivalent system, PML (promyelocytic leukemia) nuclear bodies that repair DNA damage[13], relies on the Small Ubiquitin-like Modifier (SUMO) domain that interacts with the SUMO Interacting Motif (SIM) to form droplets[14–16]. This system inspired in vitro experiments with engineered poly-SUMO and polySIM of various valences[10], and, indeed, phase separation was observed with increasing concentrations of the two polymers. Other in vitro two-component systems, e.g., an engineered polySH3-Proline-Rich Motif system[8] and a PTB-RNA system[9], also form droplets as concentrations are increased.

A striking theoretical prediction regarding these two-component multivalent systems is that, in the regime of strong binding, condensation will be extremely sensitive to the relative valence of the two components[7]. Higher valence is normally expected to boost condensate formation[8]. In strongly bound two-component systems, however, an exception occurs when the valence of one species equals or is an integral multiple of the valence of the other species. In this case, condensation is suppressed in favor of small fully bonded oligomers[7]— a "magic-number effect" reminiscent of the exact filling of atomic shells leading to the unreactive noble gases. Here, we demonstrate that this magic-number effect still occurs if all components are flexible polymers, but the effect is maximized if one of the components is rigid and compact (aka a patchy particle or patchy colloid[17]), in which case fully-bonded oligomers are more entropically favored over the condensate. Strikingly, while the magic-number effect requires rather precise global stoichiometry when all polymers are flexible, if one of the polymers is rigid the effect occurs over a broad range of stoichiometries. While many intracellular condensates are held together by weak interactions of multiple types (e.g., charged, aromatic, and hydrophobic[18], as well as pi–cation[19], and pi–pi interactions[20]) natural protein–protein, protein–RNA, or protein–peptide interactions such as SUMO-SIM[10] or synthetic interactions, e.g., based on DNA hybridization[21,22], can readily reach the strong-binding regime required to observe the magic-number effect.

## Results

### Magic-number effect for one rigid and one flexible component.
To examine the role of rigidity in the magic-number effect, we begin with two species that interact: a flexible linear polymer and a rigid shape. The flexible polymer with $n$ binding sites is denoted as $A_n$ and the rigid $4 \times 2$ shape is denoted as $B_{8R}$; each binding site

on each species is considered to be a "monomer". Previous results from modeling the Rubisco-EPYC1 system, which established the concept of the magic-number effect, can then be summarized as follows: the $A_4:B_{8R}$ system forms stable trimers (each composed of two $A_4$s and one $B_{8R}$) and thus, compared to the $A_3:B_{8R}$ or $A_5:B_{8R}$ systems, requires a substantially higher total monomer concentration for the formation of large clusters, assuming equal total stoichiometry of A and B monomers[7].

Based on these observations, we predicted that flexible polymers of length 8 together with rigid $4 \times 2$ shapes should also constitute a magic-number system, in this case based on stable heterodimers, since each constituent has 8 binding sites. To test this idea, we simulated $A_7:B_{8R}$, $A_8:B_{8R}$, and $A_9:B_{8R}$ systems (Fig. 1a–c). The heat maps of the average cluster size for different concentrations and specific bond strengths (Fig. 1g–i) define approximate phase diagrams: regions with smaller cluster sizes correspond to a homogeneous dissolved phase, while regions with larger cluster sizes correspond to an inhomogeneous condensed phase, i.e., a phase-separated regime. Confirming the visual impression that the observed large clusters arise from phase separation, rather than arising as percolation clusters from homogeneous gelation[23], we found that for strong enough specific interactions ($\geq 4k_BT$) the internal cluster density remains approximately constant above the transition, as expected for phase separation but not for clusters formed by percolation (Supplementary Fig. 4). Moreover, the dense clusters we observe are always internally connected by a network of specific bonds. We conclude that our system represents a case of gelation driven by phase separation[23,24]. Notice that in Fig. 1g–i, as specific interactions increase from zero, cluster sizes initially increase, but stop increasing above $\sim 7k_BT$. This behavior is due to saturation of the specific bonds, i.e., for large enough specific bond energies essentially all possible bonds are formed.

As anticipated, the phase boundary for the magic-number system $A_8:B_{8R}$ occurs at substantially higher concentration than for the non-magic-number systems $A_7:B_{8R}$ or $A_9:B_{8R}$, reflecting the stability of dimers composed of one flexible $A_8$ and one rigid $B_{8R}$ polymer (Fig. 1g–i,m). Note that we define a "dimer" as consisting of one polymer of each species which only form specific bonds with each other. By inspection, the specific interaction energy required for the onset of the magic-number effect is $\sim 5k_BT$ for our simulations with components of valence $\sim 8$.

What drives phase separation in these two-component systems? In the fully bonded regime, the transition is not driven by a competition between entropy and energy as in Flory–Huggins theory[25], but rather by a competition between different types of entropy. For example, in the droplet phase of the $A_8:B_{8R}$ system, free dimers coexist with large gel-like clusters. Each dimer has high translational entropy, while each component within a cluster has very limited translational entropy. However, this difference in translational entropies is offset by an opposing difference in conformational entropies: in each dimer, the binding sites of one species must match all the binding sites of the other species, leading to a reduced total conformational entropy. By contrast, the components in a large condensed cluster are more independent, binding to multiple members of the other species and enjoying a relatively high conformational entropy. Because the importance of translational entropy relative to conformational entropy is reduced as concentration increases, the system transitions from a uniform dimer phase to a droplet phase with increasing concentration. By contrast, in non-magic-number systems (e.g., $A_7:B_{8R}$ or $A_9:B_{8R}$), the polymers cannot form free dimers without unsatisfied binding sites, and instead tend to form large fully-bonded clusters even at low concentrations. This key difference

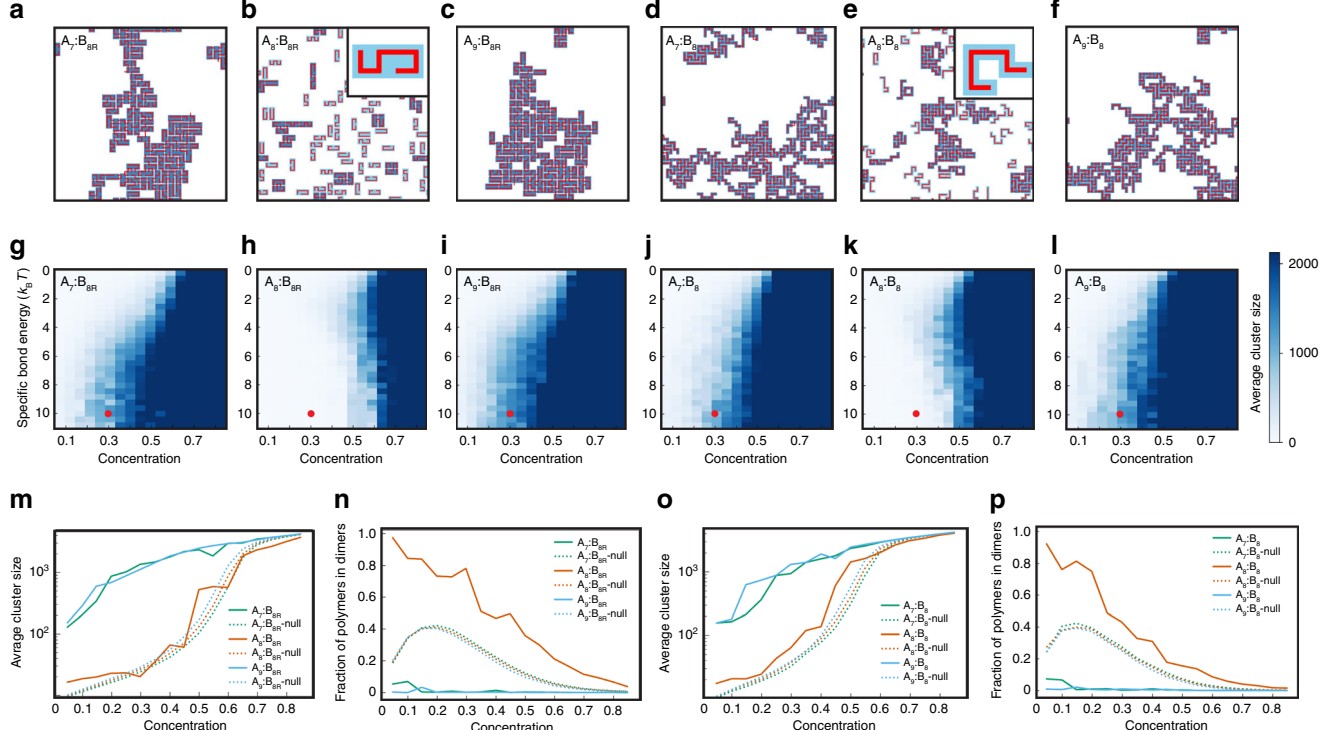

**Fig. 1 Simulations of two-component multivalent systems reveal a magic-number effect. a–f** Snapshots of 2D simulations of **a** $A_7$:$B_{8R}$, **b** $A_8$:$B_{8R}$, **c** $A_9$:$B_{8R}$, **d** $A_7$:$B_8$, **e** $A_8$:$B_8$, and **f** $A_9$:$B_8$ systems, where $A_n$/$B_n$ denotes flexible polymers of species A/B with valence $n$, and $B_{8R}$ denotes rigid $4 \times 2$ rectangles. Parameters: specific bond energy $= 10k_BT$, non-specific bond energy $= 0.1k_BT$, A:B monomer ratio $= 1$, monomer concentration $= 0.3$. Insets for the magic-number systems in **b** and **e** show characteristic fully-bonded dimers composed of one A and one B polymer. **g–l** Heat maps of average cluster size as functions of total monomer concentration and strength of specific bonds for systems in **a–f**. The ratio of A:B monomer concentration is equal to one, and the non-specific bond energy $= 0.1k_BT$. Red dots indicate parameters of snapshots in **a–f**. **m** Average cluster size for $A_7$:$B_{8R}$, $A_8$:$B_{8R}$, $A_9$:$B_{8R}$ systems (solid curves) with specific bond energy $10k_BT$, i.e., horizontal cuts through **g–i**. **n** Fraction of polymers in dimers for $A_7$:$B_{8R}$, $A_8$:$B_{8R}$, $A_9$:$B_{8R}$ systems (solid curves) with specific bond energy $10k_BT$. **o** Average cluster size for $A_7$:$B_8$, $A_8$:$B_8$, $A_9$:$B_8$ systems (solid curves) with specific bond energy $10k_BT$, i.e., horizontal cuts through **j–l**. **p** Fraction of polymers in dimers, i.e., one A-polymer and one B-polymer forming specific bonds only with each other, for $A_7$:$B_8$, $A_8$:$B_8$, $A_9$:$B_8$ systems (solid curves) with specific bond energy $10k_BT$. In **m–p**, dotted curves show results for a zero-interaction-energy null model.

leads to the drastic difference in clustering between magic-number and non-magic-number systems.

While the simulations in Fig. 1 are performed in two dimensions, because the magic-number effect depends only on the ability to form small fully bonded oligomers, it also occurs in three dimensions (Supplementary Fig. 3). In three dimensions, we observe the onset of the magic-number effect around a specific interaction energy of $4k_BT$ for components of valence $\sim 8$. We note that in three dimensions, phase separating systems can become trapped in a state of interconnected "fibers", but we confirmed that our annealing procedure leads instead to a stable droplet phase (Supplementary Fig. 5).

To mimic weak attractive interactions such as hydrophobicity, we include a small non-specific bond energy in all our simulations, which leads to more compact droplets. However, these non-specific interactions are neither sufficient nor necessary for the magic-number effect, though their magnitude influences the location of the phase boundary (Supplementary Figs. 6 and 7).

To verify that in the $A_8$:$B_{8R}$ magic-number system the phase transition in the fully-bonded regime is driven by a competition between dimers and a condensed phase, we measured the fraction of polymers in the dimer form (Fig. 1n). For magic-number systems in the low concentration regime, the polymers are essentially all in dimers, while at very high concentration the fraction of dimers is low since the system is dominated by a condensate. Notably, the fraction of dimers is always higher than in a null model (see Supplementary Note 1 for details). This

confirms the picture of an entropy-driven phase transition in which the $A_8$:$B_{8R}$ magic-number system is dominated by dimers at low concentration and a condensed phase at high concentration. By contrast, the $A_7$:$B_{8R}$ and $A_9$:$B_{8R}$ non-magic-number systems never have a high dimer fraction (Fig. 1n). Even at low concentrations, the polymers tend to aggregate into clusters to form as many bonds as possible.

**Magic-number effect is weaker if both components flexible.** In the above, we considered one flexible component and one rigid component. In principle, the magic-number effect should still occur if both components are flexible. So does the rigidity of one component matter? To answer this question, we simulated systems of two flexible polymer species, where one species has 7, 8, or 9 binding sites and the other has 8 binding sites (systems denoted as $A_7$:$B_8$, $A_8$:$B_8$, and $A_9$:$B_8$). Snapshots reveal fewer clusters in $A_8$:$B_8$ than in $A_7$:$B_8$ or $A_9$:$B_8$ (Fig. 1d–f), and heat maps of average cluster size confirm that higher concentrations are required for clustering in the $A_8$:$B_8$ system (Fig. 1j–l). Correspondingly, the $A_8$:$B_8$ system has smaller average cluster sizes (Fig. 1o) and a higher fraction of dimers (Fig. 1p) than the non-magic number systems in the strong-interaction limit. However, compared to the case with one rigid component the magic-number effect is noticeably weaker when both polymers are flexible: the onset of phase separation occurs at a lower concentration and average cluster sizes are in general larger. What is the origin of this difference?

**Polymer-dimer conformational entropy influences clustering.** A possible reason for the enhanced magic-number effect in systems with one rigid component lies in the number of distinct ways of forming dimers. Given the $4 \times 2$ rectangular shape of $B_{8R}$, an $A_8$ polymer has 28 different ways to pack inside the rectangle, i.e., the dimer degeneracy is $4 \times 28 = 112$, where the factor of 4 comes from the four possible square-lattice orientations of a rectangle with a defined "head" site. By comparison, the dimer degeneracy for $A_8 : B_8$ is higher, since there are 9960 distinct ways for two length $L = 8$ polymers to occupy the same lattice sites, with the head of one of the polymers at a defined site. However, to gauge the effect of dimer degeneracy on clustering, one should actually compare the dimer degeneracy to the number of possible conformations of the two polymers considered separately, since the two polymers can sample conformations independently within a condensed cluster. For the $A_8 : B_{8R}$ system, the number of independent conformations is $4 \times 2172$, since 2172 is the number of conformations of a polymer of length $L = 8$ with a defined head site; correspondingly, for the $A_8 : B_8$ system the number of independent conformations is $2172 \times 2172$. Therefore, the fold-reduction in the number of conformations upon dimerization in the semi-rigid $A_8 : B_{8R}$ system is $(4 \times 2172)/112 = 78$, while the fold-reduction for the flexible $A_8 : B_8$ system is substantially larger $(2172 \times 2172)/9960 = 474$. The greater loss of conformations upon dimerization in the fully flexible $A_8 : B_8$ system will favor clusters over dimers, and could therefore explain the larger average cluster size for $A_8 : B_8$.

To test this idea, we designed a system with a minimal dimer degeneracy, but otherwise as similar as possible to the semi-rigid $A_8 : B_{8R}$ system. Namely, we considered a rigid $4 \times 2$ shape together with a flexible polymer, but with a rule that allows only a single "U-shaped" conformation of the flexible polymer to achieve the full dimer binding energy, while still permitting high conformational entropy of the flexible polymers within large clusters (see Methods). Dimers in this $A_8 : B_{8U}$ system have an effective degeneracy of only $4 \times 2 = 8$. As shown in Fig. 2a,b, snapshots of simulations of the $A_8 : B_{8U}$ system with monomer concentration 0.4 (i.e., 40% of the lattice sites occupied by each species) dramatically confirm that lower dimer degeneracy leads to more clustering. The $A_8 : B_{8R}$ system with dimer degeneracy 112 is mostly composed of dimers, with only a few small clusters, while the $A_8 : B_{8U}$ system with dimer degeneracy 8 is dominated by one large cluster. Averaged results obtained at different concentrations (Fig. 2c) confirm that lower dimer degeneracy leads to more clustering.

**Relative concentration of monomers influences clustering.** So far, we have considered cases where the two polymer species have the same total number of monomers in the simulation domain, i.e., the same monomer concentration. How do differences in monomer concentration influence clustering for magic-number and non-magic-number systems? To address this question, we compared the systems $A_7 : B_8$ to $A_8 : B_8$ (both polymers flexible) and $A_7 : B_{8R}$ to $A_8 : B_{8R}$ (one flexible polymer and one rigid $4 \times 2$ shape) over a range of relative monomer concentrations. To avoid the confounding effect of changing the total monomer concentration, we fixed the sum of the concentrations of monomers of the two species, while varying the ratio of the two concentrations from 1:2 to 2:1 (Fig. 3).

Both non-magic-number cases, $A_7 : B_8$ and $A_7 : B_{8R}$, display a peak of average cluster size around equal monomer concentration. This is because at equal concentration there are no excess monomers of either species, so the system is driven to form large clusters in order to maximize the total number of specific bonds.

By contrast, the magic-number system $A_8 : B_8$ with both polymers flexible displays a sharp dip in average cluster size

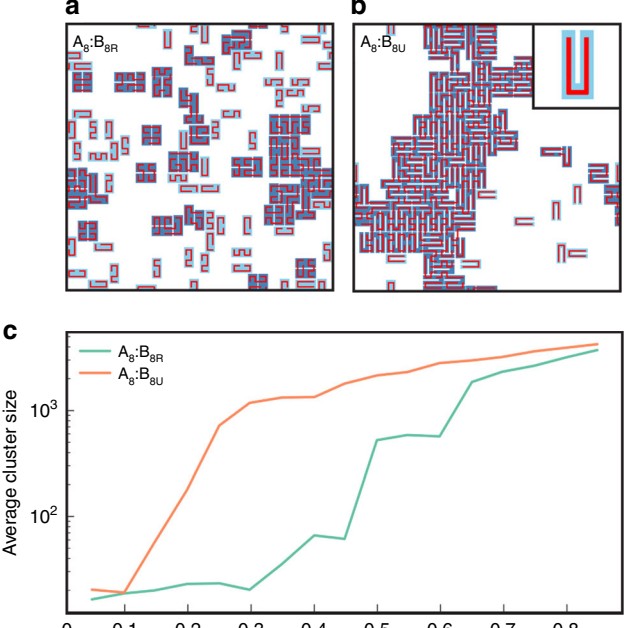

**Fig. 2 Lower conformational entropy of dimers favors clustering.** Snapshots of simulations of **a** $A_8 : B_{8R}$, and **b** $A_8 : B_{8U}$ systems. $B_{8U}$ denotes rigid $4 \times 2$ rectangles that restrict a dimerizing polymer partner to adopt a specific U-shape (see Methods). Parameters: specific bond energy $= 10 k_B T$, non-specific bond energy $= 0.1 k_B T$, A:B monomer ratio $= 1$, monomer concentration $= 0.4$. Inset in **b** shows a fully-bonded U-shaped dimer. **c** Average cluster size versus monomer concentration for systems in **a**, **b**.

around equal monomer concentration. For strong specific interactions, equal monomer concentration implies that all monomers are in specific bonds. In this case, for the concentration of 0.3 shown in Fig. 3a, entropic considerations favor dimers over clusters. However, away from equal concentrations the resulting excess of monomers of one polymer species greatly increases the internal conformational entropy of clusters, leading to a rapid increase of average cluster size. The average cluster size is symmetric around equal monomer concentration because the $A_8$ and $B_8$ polymers are equivalent.

Strikingly, as shown in Fig. 3b, the average cluster size in the $A_8 : B_{8R}$ system with one flexible polymer and one rigid shape is asymmetric around equal concentration. Specifically, the system with an excess of the rigid shape $B_{8R}$ (left of center) is much more clustered than the system with an excess of the flexible polymer $A_8$ (right of center). An intuitive argument explains this asymmetry: adding an excess rigid shape to a cluster allows many new configurations of the flexible polymers, while adding an excess flexible polymer to a cluster does not allow many new configurations of the rigid shapes. Importantly, this asymmetry implies a robustness of the magic-number effect for systems in which flexible polymers interact with more compact multivalent objects; specifically, a magic-number ratio of binding sites disfavors clustering—with no fine tuning of concentration—provided the flexible polymer is in excess.

**Mean-field theory for the strong interaction limit.** Can we go beyond simulations to better understand the role of rigidity in the magic-number effect? The well-known Flory–Huggins theory[25] models the competition between mixing entropy, which favors a dissolved phase, and interactions, which typically favor phase separation. While successful in explaining phase separation in

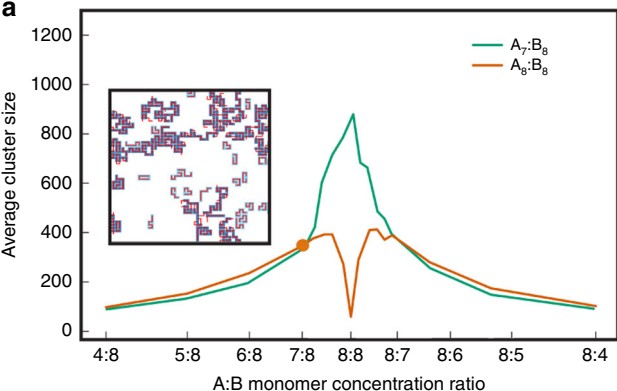

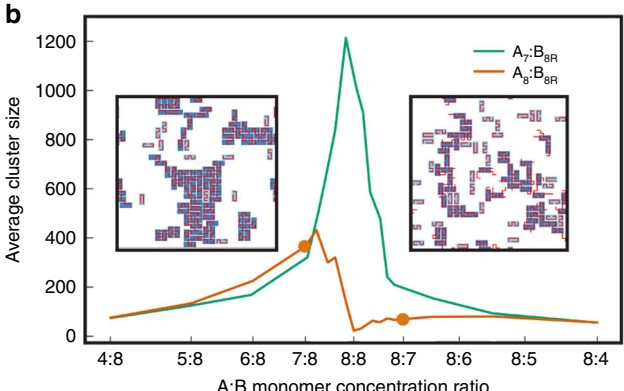

**Fig. 3 Relative concentration of monomers strongly influences clustering. a** Average cluster size of $A_7:B_8$ and $A_8:B_8$ systems versus monomer ratio A:B, for average concentration 0.3. Inset: snapshot of the $A_8:B_8$ system at monomer ratio 7:8, indicated by orange dot. **b** Average cluster size of $A_7:B_{8R}$ and $A_8:B_{8R}$ systems versus monomer ratio A:B, for average concentration 0.3. Parameters: specific bond energy $= 10k_BT$, non-specific bond energy $= 0.1k_BT$. Insets: snapshots of the $A_8:B_{8R}$ system at monomer ratios 7:8 and 8:7, indicated by orange dots.

polymer solutions, Flory–Huggins theory only considers non-specific interactions. More recently, Seminov and Rubinstein[24] presented a sticker-solution theory that incorporates specific one-to-one interactions among "stickers" on a single polymer species. The one-component sticker theory predicts both a gelation transition and condensate formation. However, two-component multivalent systems with specific interactions present additional mechanisms and a wider range of behaviors. Notably, as described above, in the limit of high binding energies in these systems the internal energy is constant—essentially all specific bonds are formed—so the phase transition is a purely entropic effect driven by a competition between a dimer (or small oligomer) phase with higher translational entropy and a condensate phase with higher conformational entropy[7].

To analyze the role of rigidity in the regime of high binding energies, we developed a simple theoretical model for our lattice-based polymer system. Specifically, we consider two species of lattice polymers with equal monomer concentrations in the strong interaction limit. Thus every monomer is in a specific bond.

For one-component lattice-polymer systems, Flory–Huggins theory provides a theoretical framework to compute the configurational entropy and internal energy, and thus the free energy[25]. For a two-component system in the fully-bonded regime, the total internal energy is a constant, and so we focus on the configurational entropy. We approximate the configurational entropy in two limits: the dilute limit dominated by dimers or

other oligomers and the dense limit dominated by a condensate (see Supplementary Note 1 for details).

We start by considering two magic-number systems composed of flexible polymers and rigid shapes, but with different dimer degeneracies: $D_{8R} = 4 \times 28 = 112$ for $A_8:B_{8R}$ and $D_{8U} = 4 \times 2 = 8$ for $A_8:B_{8U}$. In the dilute limit, we approximate the system as all dimers, in which case the combined translational entropy and internal conformational entropy of the dimers yields a free-energy density

$$f_{\text{dilute}} = \frac{c}{L}\log\left(\frac{c}{L}\right) + (1-c)\log(1-c) \\ + \frac{c}{L}(L-1) - \frac{c}{L}\log(D_{8R/U}), \quad (1)$$

where $c$ is the monomer concentration of each species and $L$ is the polymer length, and the last term captures the dependence on dimer degeneracy.

In the dense limit, we treat the two species as independently adopting all possible configurations on the lattice, but then apply a correction that reflects the requirement that the occupied sites perfectly overlap, which in a mean-field approximation yields

$$f_{\text{dense}} = \frac{2c}{L}\log\left(\frac{c}{L}\right) - c\log(c) + (1-c)\log(1-c) \\ + \frac{2c}{L}(L-1) - \frac{c}{L}[\log(C_8) + \log(C_{8R})], \quad (2)$$

where $C_8 = 2172$ is the number of conformations of a flexible polymer of length $L = 8$ with a defined head site, while $C_{8R} = 4$ is the corresponding number of conformations for the rigid shape.

In the thermodynamic limit, the lower of the two free energies dominates. The non-convexity of the resulting free-energy density as a function of concentration implies a region of coexistence of a dilute and a dense phase, and the phase boundaries can be determined from the convex hull of the free energy (Supplementary Fig. 8).

Varying the dimer degeneracy changes the free energy per dimer, captured by the last term in Eq. (1) and thus influences the phase boundary. The simple theory predicts a concentration of 0.61 for the phase transition of $A_8:B_{8U}$ with its dimer degeneracy of 8, while predicting that $A_8:B_{8R}$ with dimer degeneracy 112 never forms condensates. While the theory is too simplified to be quantitative, it qualitatively correctly captures the pronounced shift to higher concentration of the $A_8:B_{8R}$ phase boundary because of its higher dimer degeneracy.

The above model can be readily adapted to systems with two flexible polymers each of length $L$, and provides insight into the dependence of the magic-number effect on valence (see Supplementary Note 1). We continue to assume equal concentration of monomers and the strong interaction limit, in which case the free energy in the dilute limit is given by Eq. (1) with dimer degeneracy $D_L$, and in the dense limit by Eq. (2), with both conformation numbers replaced by $C_8$.

For the non-magic number system $A_{n-1}:B_n$ in the dense limit, a version of Eq. (2) that takes into account the different polymer lengths still holds (Eq. S25). In the dilute limit, the smallest fully-bonded oligomer includes $n(n-1)$ monomers of each type. We roughly approximate the oligomer conformational degeneracy by assuming the oligomer adopts an $n \times (n-1)$-rectangular shape to obtain a dilute-limit free-energy density (Eq. S28). Using these free energies to compare the magic-number systems $A_n:B_n$ and the non-magic-number systems $A_{n-1}:B_n$, the predicted phase boundary decreases from 0.68 to 0.15 for $n = 4$, from 0.72 to 0.02 for $n = 6$, and from 0.76 to 0.01 for $n = 8$. Thus, this simple theory captures the trend that the phase boundary decreases more, i.e., the magic-number effect is stronger, for higher-valence systems (see Supplementary Note 1 and Supplementary Fig. 2).

## Discussion

Motivated by the key roles played by membrane-free organelles in a variety of cellular functions, we studied phase-separable systems composed of two species of multivalent polymers that form one-to-one specific bonds. In particular, motivated by the Rubisco-EPYC1 system of the algal pyrenoid, we focused on the role of rigidity of one of the multivalent components. The results reported here are based on simple lattice polymer models supported by equally simple analytical theory; nevertheless, the model and theory capture essential features of real systems: (i) There is a phase transition from a uniform solution to liquid droplets. (ii) The low concentration phase is dominated by small oligomers. (iii) Strong one-to-one specific interactions can lead to a magic-number effect that implies striking exceptions to the general rule that higher valence favors condensation. This last feature is a key difference with respect to standard phase-separating systems, e.g., those described by Flory–Huggins theory, in which interactions are typically weak, non-specific, and non-saturable, and for which there is no magic-number effect. (Previous work on two-component systems with strong one-to-one binding[26,27] adopted the no-cycles or tree approximation, which allows at most one bond between any pair of molecules and thus does not capture the magic-number effect considered here.) We found that rigidity enhances the magic-number effect by increasing the relative conformational entropy of the small oligomers. Our lattice simulations are intended to provide conceptual insight, not to capture the details of polymer shapes, sizes, range of interactions, or entanglement. Nonetheless, we expect the magic-number effect and our conclusions regarding the role of rigidity to be robust with respect to these considerations, and the effect has been verified in 3D lattice (Supplementary Fig. 3) and off-lattice simulations[7].

There remain open theoretical questions. What is the nature of the condensed phase without attractive non-specific interactions or when such interactions are repulsive? While the magic-number effect persists in the absence of attractive non-specific interactions (Supplementary Fig. 7), the droplets in such systems have very low surface tension, leading to rough interfaces. How is the phase behavior affected by more general magic-number relations, e.g., one polymer species with valence $n$ and two partnering species with valences $p$ and $q$, with $n = p + q$, in particular when one or more of these species is rigid or branched[17]? One simplification of our lattice models is that the spacing between binding sites is constant, and identical for both components. Systems of real polymers or patchy particles may have less well-matched spacings between binding sites, and the chemical properties of the linkers (e.g., hydrophobicity)[28] may also influence both oligomer and cluster formation. Indeed, as shown by Harmon et al.[23] linkers with positive effective solvation volumes suppress phase separation and gelation and would thus add to the magic-number effect. Another question is whether a variant of the magic-number effect applies to polymers with opposite charges. Borgia et al.[29] speculated that the lack of phase separation in the H1-Protα system could be due in part to the nearly equal and opposite charges of the two flexible proteins, an approximate "magic-number" condition that would favor dimerization over condensation. Moreover, the observed high conformational entropy of H1-Protα dimers implies a low entropic cost of dimerization, which is the same reason a rigid component favors dimerization. Future work employing off-lattice models will address these questions, as well as the dynamical properties of multivalent, multicomponent systems.

Consideration of phase-separating systems that rely on specific interactions has strong biological motivation. Multivalent systems with specific interactions allow for "orthogonal" phase separated droplets to form: the specific interactions holding together one class of droplets will typically not interfere with those holding together another class[30]. Given the large number of distinct condensates now recognized within cells[31], droplet orthogonality is a key consideration. While the interactions that drive phase-separation in many of these systems may be weak, in principle protein-protein, protein-RNA, and RNA-RNA interactions can be strong enough to lead to magic-number effects. The required energy scale of $\sim$ 4-5$k_B T$ in our 3D simulations (Supplementary Fig. 3) can be converted to a $K_d$ value via the relation $E/k_B T = \ln(K_d \times \text{lattice site volume})$, where $E < 0$: taking a lattice site volume of 20 nm$^3$ roughly appropriate for a SUMO "monomer" of $\sim$ 90 amino-acid residues[10,32], yields a range of values $K_d \sim$ 1–2.5 mM, whereas the measured $K_d$ for SUMO and SIM monomers was 10 μM[10]. Thus for systems as strongly interacting as SUMO-SIM, magic-number effects in principle allow for mechanisms of regulation. For example, chemical modification of the effective valence of one component to change into or out of a magic-number condition has been proposed as a possible means of condensate regulation[7]. From an evolutionary perspective, magic-number conditions could either be exploited by cells, or avoided if there is selective pressure for phase separation at lower concentrations.

The magic-number effect makes definite experimental predictions. First, comparable magic-number and non-magic number systems will have very different phase boundaries. Second, suppression of phase separation by the magic-number effect is enhanced by higher valence and by higher dimer or small oligomer conformational entropy. Third, deviations from equal monomer concentration reduce the effect, but with a notable exception if one species is rigid and the flexible species is in excess. We hope that the results and predictions presented here will stimulate exploration of magic-number effects in multivalent, multicomponent systems in both natural and synthetic contexts, such as lens crystallin proteins[33] or DNA origami[34].

## Methods

**Model**. Simulations were performed using a square grid system of 50 × 50 grid points (or "sites") with periodic boundary conditions. In the model, polymers with different shapes occupy several connected (nearest neighbor) sites such that each monomer occupies one site. There are two species of polymer in each simulation, denoted as $A_n$ or $B_n$, with $n$ being the number of monomers in one polymer. A monomer of A and a monomer of B form a specific bond when they occupy the same site in the 2D lattice; no more than one A-monomer and no more than one B-monomer can occupy a site.

Some polymers are considered to be "flexible" in which case any configuration of connected nearest-neighbor sites is allowed. We also consider cases where the B-polymer ($B_{8R}$) is a rigid 4 × 2 rectangle, and a "U-shaped" variant ($B_{8U}$) described below.

Systems with only these specific interactions have a very weak effective surface tension between condensed and dilute phases, which prevents formation of dense droplets. Motivated by the existence of weak non-specific interactions between polymers (e.g., due to hydrophobicity), we add a small non-specific interaction between all nearest neighbor monomers as described below, which increases the surface tension between phases and results in denser droplets.

We performed Markov–Chain Monte–Carlo simulations using the Metropolis algorithm[35]. Briefly, in each simulation step we randomly propose a move of the configuration. The move is always accepted if it reduces system energy, and accepted with probability $e^{-(E_f - E_i)/k_B T}$, where $E_f$ and $E_i$ are the final and initial energies, if the move increases system energy. Three categories of moves are proposed: single-flexible-polymer moves, single-rigid moves, and two-species joint moves. Single-flexible-polymer moves are standard lattice-polymer local moves: the end-point move, the corner move, and the reptation move[7]. Single-rigid moves consist of one-step translations in the four cardinal directions and a 90-degree rotation around the center of the rigid shape. In the regime of strong specific bonds, the two species are typically held together by multiple specific bonds, which leads to dynamical freezing. This is more severe if one of the species is rigid, since the moves affect more binding sites. To enable the system to better explore configuration space, in the case that one species is rigid, we include two-species joint moves such that connected clusters of polymers move together, without breaking any specific bonds. The joint moves consist of translating a connected cluster of the two species of polymers together or rotating the whole cluster by 90-degrees around any point. To obtain thermalized ensembles, we follow a two-step simulated annealing procedure: we keep $k_B T$ constant and gradually increase bond strength. We first increase the non-specific bond from 0 to 0.1$k_B T$ in 0.005$k_B T$ increments, keeping the specific

bond energy at $0k_BT$. Then the specific bond energy is increased from 0 to $11k_BT$ in $0.04k_BT$ increments, while the non-specific bond energy is kept at $0.1k_BT$. Each step of annealing is simulated with at least 50,000 Monte–Carlo steps (i.e., proposed moves) per monomer to ensure complete thermalization and results are averaged over 20-100 of the resulting thermalized snapshots.

**Construction of the "U-shaped" $A_8:B_{8U}$ system.** To test how the phase diagram is affected by dimer degeneracy, we designed a variant of the $A_8:B_{8R}$ system that minimizes the number of possible dimer conformations. Specifically, we defined a "U-shaped" variant of the $4 \times 2$ rectangle ($B_{8U}$), such that an A-polymer crossing the central long axis of the rectangle (except at one end) only contributes one binding energy from the two adjacent binding sites. This energetically favors A-polymer partners that follow the U-shape (Supplementary Fig. 1).

**Reporting summary.** Further information on research design is available in the Nature Research Reporting Summary linked to this article.

## Data availability

Data supporting the findings of this manuscript are available from the corresponding author upon reasonable request. A reporting summary for this Article is available as a Supplementary Information file.

## Code availability

The code used in this study is available at https://github.com/binarybin/polymersim (for two-dimensional simulations) and https://github.com/BenjaminWeiner/magic-numbers (for three-dimensional simulations).

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

## Acknowledgements

We thank Farzan Beroz, Xiaowen Chen, Amir Erez, Shan He, and Zhiyuan Li for insightful discussions and encouragement. This work was supported in part by the NSF, through the Center for the Physics of Biological Function (PHY-1734030), and through Grant IOS-1359682 (M.C.J.), and by the NIH under Grant No. 7DP2GM119137-02 (M.C.J.). G.H. was supported by a China Scholarship Council scholarship. P.R. was supported by a Princeton Center for Theoretical Science fellowship. B.W. was supported by a Joseph H. Taylor Graduate Student Fellowship.

## Author contributions

B.X. and N.S.W. conceived the research. B.X., G.H., and B.G.W. performed and analyzed simulations. B.X., Y.M., P.R., and N.S.W. developed the analytical model. All authors contributed to interpreting results and preparing the manuscript.

## Competing interests

The authors declare no competing interests.
