## [Peer Review File · Nature Communications]

Reviewers' Comments:

Reviewer #1:

Remarks to the Author:

The revised submission is vastly improved over the version that was submitted to Nature Physics. The key takeaway here is that valence alone is an insufficient arbiter of phase separation if the one-to-one interactions are stronger than a certain threshold. In this scenario, the onset of phase separation is governed by whether or not the system is in the magic number regime - where all bonds can be saturated - or outside the magic number regime. In the magic number regime, 1:1 complexes, i.e., dimers can suppress both percolation (aka gelation) and phase separation. I've been puzzled by this observation and given previous simulations (see Ref. 23) the question is if the signatures of the magic number effect were always present. The revised version, which is a lot clearer than the original submission, made it very easy to understand the core of the message. And indeed, there is a direct correspondence between the findings in Ref. 23 and the current work. Please see Fig. 8 in Ref. 23. Here, the authors compute C^* , which is the ratio of the percolation threshold calculated using their simulations to the percolation threshold calculated using Flory-Stockmayer theory. When $C^* > 1$, the implication is that phase separation does not occur and that in fact, percolation is also suppressed. For a 3-3, 5-5, and 7-7 mimic of the SH3-PRM system, the authors find that $C^* > 1$ for short linkers, which means increasing contributions from the rigid SH3 and PRM domains. This is congruent with the current work. What is interesting is that for linkers of high excluded volume, even for long linkers, percolation is suppressed, see panel (b) in Figure 8 of Ref. 23. This implies that the magic number effect is amplified by the properties of linkers, specifically their excluded volumes, so much so that even non-magic number systems with higher excluded volume linkers can suppress phase transitions, again due to entropy, except the entropy here is the solvation entropy.

I bring this point up because it actually suggests that the union of the authors's work and that of Ref. 23 provide a rather complete picture of what might be going on in systems with strong 1:1 interactions. This would be worth mentioning.

It is also worth noting that it would be erroneous to conflate flexibility with weak and non-specific interactions. This, as the Schuler lab has shown in their 2018 Nature paper, does not have to follow. In fact, the interaction strengths there are considerably stronger than the SH3:PRM, SUMO:SIM and other interactions. Indeed, this system i.e., the H1a-ProT-alpha system, does not undergo phase separation via complex coacervation and recent data from the Schuler lab suggests that phase separation can only happen when the strong interactions are screened. I believe the Schuler work should be cited and discussed because it does challenge the magic number and rigid vs. flexible idea to some extent.

Other than these two revisions, I have no further changes to recommend and believe that a suitably revised version would be perfect for publication in Nature Communications.

Reviewer #2:

Remarks to the Author:

I have reviewed the article by Xu and colleagues for a different journal previously. The new manuscript now includes a 3D model and includes non-specific interactions. The text has also been carefully revised to address my earlier concern that the simulated results are not fully consistent with published experimental data on the EPYC1-Rubisco system.

In summary the work has been significantly expanded and revised. My earlier comments have been addressed adequately. The paper represents a thorough theoretical development of the Magic Numbers hypothesis. I look forward to future reports that experimentally test the concept using approaches such as biochemical reconstitution.

Reviewer #3:

Remarks to the Author:

Based on the Rubisco and EPYC1 system involved in the formation of the algal pyrenoid, the authors build a lattice model to simulate how valencies of multiple modular domains of the two proteins influence their phase separation behavior. They found that when the attractive interaction between the two proteins is strong and the valencies of the two proteins are identical, the proteins tend to form independent heterodimers and only phase separate when their concentrations are very high, a phenomenon called the "magic-number effect". They argue that phase separation in such a system occurring with strong attractive interactions and high concentration is a result of a competition between translational entropy (bigger in the dimer phase but lower in the droplet phase) and conformational entropy (bigger in droplet phase but lower in dimer phase), instead of a competition between entropy and enthalpy. They simulate two different systems, one with two flexible polymers and the other with one rigid scaffold and one flexible polymer, and found that the magic-number effect is stronger in the rigid-plus-flexible system.

The authors discuss an interesting potential mechanism for regulation of phase separation in scaffold-substrate enzyme systems containing a rigid component, which is different from the more common picture that describes the condensation of two more flexible proteins (multi-domain proteins with significant disordered regions). The results are intriguing and are of interest to the phase separation field but there are some concerns.

(1) A major concern is with the generality of the magic-number effect and its biological relevance. The authors emphasize in the Introduction that they are exploring "the regime of strong binding" where they claim that the magic-number effect will dominate. In the Discussion the energy scale of this "regime of strong binding" is given as $> 4\sim 5kT$ corresponding to $K_d \sim 1mM$. With the example of SUMO-SIM binding affinity of $K_d \sim 10 \mu M \ll 1mM$, the authors argue that real biological systems can exist in the strong-binding regime and thus magic-number effects "in principle allow for novel mechanisms of regulation". However, there are no experimental validations of this effect for SUMO-SIM or other modular binding domain-motif systems. The authors do not provide any experimental system or experimental data as an example of a case obeying the magic-number effect.

In fact, phase separation has been shown to be triggered by increased concentration of pairs of proteins with an identical number of attractive modular binding domains for SUMO-SIM and SH3-proline-rich-motif systems, as well as for PTB-RNA, as noted in the Introduction. Simulation also shows that a solution of a pair of multiple modular domain proteins with identical valency will undergo phase separation and gelation [Harmon et al (2017) eLife 6 e30294; Ref 23 of this manuscript], but not the dimerization suggested by the magic-number effect. The authors can argue that these systems do not have strong enough binding energy, and thus do not have the magic-number effect, as they have stated in the Supplementary Information that "There is no magic-number effect for weak interactions". However, given such known counter-examples, the authors must acknowledge the much

more common case of weak interactions in biological phase separation, instead of burying this in the supplement. They should clarify the limitations and valid scope of their theory, including the seemingly dominant role of either weak interactions among disordered protein regions and/or nucleic acids or among modular binding domains for their cognate motifs with each surrounded by disordered linker regions. Given that a third of the proteome has significant stretches of disorder, and most proteins with modular binding domains have disordered linkers, the dominant case in biology seems to be weak interactions in a flexible context.

It is very likely that the propensity for phase separation is due to a more complex solvation effect based on the sequences of disordered regions as well as the multivalent modular binding interactions, rather than or possibly in addition to the magic number. The authors should acknowledge that there are other important contributions to determining whether a system will phase separate or not.

Perhaps one biological relevance is this: in order to optimize the cell's ability to exploit phase separation, disorder and weak interactions have been exploited. The authors could also identify other rigid scaffold systems that may behave subject to the magic-number effect.

(2) More technical: Some of the authors' claims in the section of "Phase separation versus percolation" in the supplement (p.6) are not obvious:

(i) The authors claim that "Within our mean-field theory, the system clearly undergoes phase separation: the free energy is non-convex and the order parameter (density) has a discontinuity, both hallmarks of a first-order phase transition." (line 94-95). First, the authors do not show any free energy profile to support their claim that "the free energy is non-convex"; second, the authors do not explain why solute density can be a good order parameter for phase separation; other quantities, e.g. cluster size, can also be an order parameter. Third, even if the above two points are satisfied, the logical link between "hallmarks of a first-order phase transition" and "clearly undergoes phase separation" is not clear and this requires an explanation.

(ii) The authors argue that their system does not involve gelation by stating that "In the Monte Carlo simulations, local density analysis suggests that the macroscopic clusters are phase-separated droplets rather than percolation clusters arising from random overlaps between polymers." (line 95-97). It seems that the authors are not aware that gelation and phase separation can occur simultaneously in a system: as gelation is driven by intermolecular interaction (between the "stickers"), this interaction can also trigger phase separation; on the other hand, phase separation can create a condensed phase where concentration is high enough to trigger gelation. The interplay between gelation and phase separation has been greatly explored in the previously mentioned Harmon et al. eLife 2017 paper. It is thus not clear how the authors' simulation supports their claim that their system does not have percolation clusters.

Response to Critiques

Reviewer #1 (Remarks to the Author):

The revised submission is vastly improved over the version that was submitted to Nature Physics. The key takeaway here is that valence alone is an insufficient arbiter of phase separation if the one-to-one interactions are stronger than a certain threshold. In this scenario, the onset of phase separation is governed by whether or not the system is in the magic number regime - where all bonds can be saturated - or outside the magic number regime. In the magic number regime, 1:1 complexes, i.e., dimers can suppress both percolation (aka gelation) and phase separation. I've been puzzled by this observation and given previous simulations (see Ref. 23) the question is if the signatures of the magic number effect were always present. The revised version, which is a lot clearer than the original submission, made it very easy to understand the core of the message. And indeed, there is a direct correspondence between the findings in Ref. 23 and the current work. Please see Fig. 8 in Ref. 23. Here, the authors compute C^* , which is the ratio of the percolation threshold calculated using their simulations to the percolation threshold calculated using Flory-Stockmayer theory. When $C^* > 1$, the implication is that phase separation does not occur and that in fact, percolation is also suppressed. For a 3-3, 5-5, and 7-7 mimic of the SH3-PRM system, the authors find that $C^* > 1$ for short linkers, which means increasing contributions from the rigid SH3 and PRM domains.

This is congruent with the current work. What is interesting is that for linkers of high excluded volume, even for long linkers, percolation is suppressed, see panel (b) in Figure 8 of Ref. 23. This implies that the magic number effect is amplified by the properties of linkers, specifically their excluded volumes, so much so that even non-magic number systems with higher excluded volume linkers can suppress phase transitions, again due to entropy, except the entropy here is the solvation entropy.

I bring this point up because it actually suggests that the union of the authors's work and that of Ref. 23 provide a rather complete picture of what might be going on in systems with strong 1:1 interactions. This would be worth mentioning.

We thank the reviewer for highlighting the relevant results regarding the role of linkers in Ref. 23. We agree that long self-avoiding linkers will augment the magic-number effect and further suppress phase separation, as well as gelation. We have acknowledged this point in the Discussion: "Indeed, as shown by Harmon et al.²³ linkers with positive effective solvation volumes suppress phase separation and gelation and would thus add to the magic-number effect."

It is also worth noting that it would be erroneous to conflate flexibility with weak and non-specific interactions. This, as the Schuler lab has shown in their 2018 Nature paper, does not have to follow. In fact, the interaction strengths there are considerably stronger than the SH3:PRM, SUMO:SIM and other interactions. Indeed, this system i.e., the H1a-ProT-alpha system, does not undergo phase separation via complex coacervation and recent data from the Schuler lab suggests that phase separation can only happen when the strong interactions are screened. I believe the Schuler work should be cited and discussed because it does challenge the magic number and rigid vs. flexible idea to some extent.

The reviewer's comment highlights an interesting relation between the cited study by Borgia et al. (Nature 2018) and the magic-number effect. Borgia et al. study the interaction of two largely disordered and strongly oppositely charged proteins, H1 and ProTx. They speculate that the strong dimerization and lack of phase separation could be due in part to the nearly equal charges of the two flexible

proteins. Furthermore, their observation of a high conformational entropy of H1-ProT α dimers is actually consistent with our observation that a rigid component favors dimerization over condensation: both cases imply a low entropic cost of dimerization. We now highlight this connection in the Discussion: "Another question is whether a variant of the magic-number effect applies to polymers with opposite charges. Borgia et al.²⁹ speculated that the lack of phase separation in the H1-ProT α system could be due in part to the nearly equal and opposite charges of the two flexible proteins, an approximate "magic-number" condition that would favor dimerization over condensation. Moreover, the observed high conformational entropy of H1-ProT α dimers implies a low entropic cost of dimerization, which is the same reason a rigid component favors dimerization." We thank the reviewer for bringing this relevant result to our attention.

Other than these two revisions, I have no further changes to recommend and believe that a suitably revised version would be perfect for publication in Nature Communications.

We are gratified by the reviewer's conclusion that a revised version of our manuscript would be suitable for publication in Nature Communications.

Reviewer #2 (Remarks to the Author):

I have reviewed the article by Xu and colleagues for a different journal previously. The new manuscript now includes a 3D model and includes non-specific interactions. The text has also been carefully revised to address my earlier concern that the simulated results are not fully consistent with published experimental data on the EPYC1-Rubisco system.

In summary the work has been significantly expanded and revised. My earlier comments have been addressed adequately. The paper represents a thorough theoretical development of the Magic Numbers hypothesis. I look forward to future reports that experimentally test the concept using approaches such as biochemical reconstitution.

We thank the reviewer for this positive assessment of our revised manuscript.

Reviewer #3 (Remarks to the Author):

Based on the Rubisco and EPYC1 system involved in the formation of the algal pyrenoid, the authors build a lattice model to simulate how valencies of multiple modular domains of the two proteins influence their phase separation behavior. They found that when the attractive interaction between the two proteins is strong and the valencies of the two proteins are identical, the proteins tend to form independent heterodimers and only phase separate when their concentrations are very high, a phenomenon called the "magic-number effect". They argue that phase separation in such a system occurring with strong attractive interactions and high concentration is a result of a competition between translational entropy (bigger in the dimer phase but lower in the droplet phase) and conformational entropy (bigger in droplet phase but lower in dimer phase), instead of a competition between entropy and enthalpy. They simulate two different systems, one with two flexible polymers and the other with one rigid scaffold and one flexible polymer, and found that the magic-number effect is stronger in the rigid-plus-flexible system.

The authors discuss an interesting potential mechanism for regulation of phase separation in scaffold-substrate enzyme systems containing a rigid component, which is different from the more common

picture that describes the condensation of two more flexible proteins (multi-domain proteins with significant disordered regions). The results are intriguing and are of interest to the phase separation field but there are some concerns.

We appreciate the reviewer's careful reading of our revised manuscript, and are pleased that he/she finds our results intriguing.

(1) A major concern is with the generality of the magic-number effect and its biological relevance. The authors emphasize in the Introduction that they are exploring "the regime of strong binding" where they claim that the magic-number effect will dominate. In the Discussion the energy scale of this "regime of strong binding" is given as $> 4\sim 5kT$ corresponding to $K_d \sim 1\text{mM}$. With the example of SUMO-SIM binding affinity of $K_d \sim 10\ \mu\text{M} \ll 1\text{mM}$, the authors argue that real biological systems can exist in the strong-binding regime and thus magic-number effects "in principle allow for novel mechanisms of regulation". However, there are no experimental validations of this effect for SUMO-SIM or other modular binding domain-motif systems. The authors do not provide any experimental system or experimental data as an example of a case obeying the magic-number effect.

We absolutely share the reviewer's interest in experimental validation and exploration of the magic number effect. As the effect occurs robustly in simulation (and indeed helps explain results of the previous theoretical work in Ref. 23 as pointed out by Reviewer #1), and may be realized in multiple systems including protein, RNA, DNA, and non-biological polymers and patchy particles, we believe that the best way to achieve this desirable goal is to communicate our current theoretical results to a broad audience via publication of our study.

In fact, phase separation has been shown to be triggered by increased concentration of pairs of proteins with an identical number of attractive modular binding domains for SUMO-SIM and SH3-proline-rich-motif systems, as well as for PTB-RNA, as noted in the Introduction. Simulation also shows that a solution of a pair of multiple modular domain proteins with identical valency will undergo phase separation and gelation [Harmon et al (2017) eLife 6 e30294; Ref 23 of this manuscript], but not the dimerization suggested by the magic-number effect.

We note that the magic-number effect does not prevent phase separation, but rather shifts the phase boundary to higher concentrations relative to non-magic-number systems. Moreover, for the case in which both components are flexible, the effect requires near equality of the concentrations of both types of monomers. Thus the observation of phase separation in the experimental studies cited by the reviewer do not contradict the effect, rather we hope that our work will inspire additional consideration of these systems in the regime where the effect is expected to occur. Similarly, the theoretical simulations cited by the reviewer do not contradict the effect, but rather highlight that the magnitude of the effect does depend on a variety of parameters such as binding energies.

The authors can argue that these systems do not have strong enough binding energy, and thus do not have the magic-number effect, as they have stated in the Supplementary Information that "There is no magic-number effect for weak interactions". However, given such known counter-examples, the authors must acknowledge the much more common case of weak interactions in biological phase separation, instead of burying this in the supplement. They should clarify the limitations and valid scope of their theory, including the seemingly dominant role of either weak interactions among disordered protein regions and/or nucleic acids or among modular binding domains for their cognate motifs with each surrounded by disordered linker regions. Given that

a third of the proteome has significant stretches of disorder, and most proteins with modular binding domains have disordered linkers, the dominant case in biology seems to be weak interactions in a flexible context.

We acknowledge the reviewer's important point that weak interactions among intrinsically disordered regions and/or modular binding domains are ubiquitous in phase-separating systems. Indeed, we already acknowledged this point in the Introduction: "While many intracellular condensates are held together by weak interactions of multiple types (e.g. charged, aromatic, and hydrophobic¹⁸, as well as pi-cation¹⁹, and pi-pi interactions²⁰)..." . We now remind the reader in the Discussion that weak interactions can mediate phase separation: "While the interactions that drive phase-separation in many of these systems may be weak, protein-protein, protein-RNA, and RNA-RNA interactions can in principle be strong enough to lead to magic-number effects."

It is very likely that the propensity for phase separation is due to a more complex solvation effect based on the sequences of disordered regions as well as the multivalent modular binding interactions, rather than or possibly in addition to the magic number. The authors should acknowledge that there are other important contributions to determining whether a system will phase separate or not.

Agreed. To acknowledge the range of factors contributing to phase separation, we have added the following sentence at the conclusion of the first paragraph: "The relevant properties of components include the presence of intrinsically disordered regions, as well as the valence, strength, and specific sequence of interacting residues or domains."

Perhaps one biological relevance is this: in order to optimize the cell's ability to exploit phase separation, disorder and weak interactions have been exploited. The authors could also identify other rigid scaffold systems that may behave subject to the magic-number effect.

This is a good suggestion. We are definitely interested in other systems, natural or synthetic, that might exhibit the magic-number effect. One relevant natural system of rigid components that exhibits liquid-liquid phase separation is the lens crystallin proteins. A relevant synthetic system is DNA origami. To highlight these examples, we have modified the final sentence of the Discussion to read: "We hope that the results and predictions presented here will stimulate exploration of magic-number effects in multivalent, multicomponent systems in both natural and synthetic contexts, such as lens crystallin proteins³³ or DNA origami³⁴."

(2) More technical: Some of the authors' claims in the section of "Phase separation versus percolation" in the supplement (p.6) are not obvious:

(i) The authors claim that "Within our mean-field theory, the system clearly undergoes phase separation: the free energy is non-convex and the order parameter (density) has a discontinuity, both hallmarks of a first-order phase transition." (line 94-95). First, the authors do not show any free energy profile to support their claim that "the free energy is non-convex"; second, the authors do not explain why solute density can be a good order parameter for phase separation; other quantities, e.g. cluster size, can also be an order parameter. Third, even if the above two points are satisfied, the logical link between "hallmarks of a first-order phase transition" and "clearly undergoes phase separation" is not clear and this requires an explanation.

This quoted statement refers to the free-energy profiles in Supplementary Figure 8. To clarify this point, we have modified the section to read “Within our mean-field theory, the system clearly undergoes phase separation as a first-order phase transition: the free energy is non-convex and the order parameter (density) has a discontinuity, both hallmarks of a first-order transition. Specifically, the non-convex profile of the free energy implies that the ground state of the system in the thermodynamic limit necessarily consists of two phases for all concentrations lying between the points where the tie line is tangent to the free-energy profile, and the densities of the two phases are those corresponding the points of tangency (cf. Supplementary Fig. 8). (Note that although we use mean cluster size as an order parameter throughout this work, here we use density because it exhibits more spatial variation in the finite lattice simulations and facilitates theoretical calculations.)”

(ii) The authors argue that their system does not involve gelation by stating that "In the Monte Carlo simulations, local density analysis suggests that the macroscopic clusters are phase-separated droplets rather than percolation clusters arising from random overlaps between polymers." (line 95-97). It seems that the authors are not aware that gelation and phase separation can occur simultaneously in a system: as gelation is driven by intermolecular interaction (between the "stickers"), this interaction can also trigger phase separation; on the other hand, phase separation can create a condensed phase where concentration is high enough to trigger gelation. The interplay between gelation and phase separation has been greatly explored in the previously mentioned Harmon et al. eLife 2017 paper. It is thus not clear how the authors' simulation supports their claim that their system does not have percolation clusters.

We absolutely agree that gelation and phase separation can occur simultaneously. Indeed, this point is already addressed in the main text where we write “Confirming the visual impression that the observed large clusters arise from phase separation, rather than arising as percolation clusters from homogeneous gelation²³, we found that for strong enough specific interactions ($> 4k_B T$) the internal cluster density remains approximately constant above the transition, as expected for phase separation but not for clusters formed by percolation (Supplementary Fig. 4). Moreover, the dense clusters we observe are always internally connected by a network of specific bonds. We conclude that our system represents a case of gelation driven by phase separation^{23,24}.” The latter reference is the Harmon et al. eLife 2017 paper referred to by the reviewer. To reiterate this point in the section of Supplementary Material cited by the reviewer, we have modified the quoted passage to read “In the Monte Carlo simulations, local density analysis suggests that the macroscopic clusters are phase-separated droplets rather than percolation clusters arising from homogeneous gelation.” And we conclude the paragraph with: “As the macroscopic clusters we observe are always internally connected by a network of specific bonds, we conclude that our system represents a case of gelation driven by phase separation^{5,6}.”

Reviewers' Comments:

Reviewer #3:

Remarks to the Author:

The authors have addressed our previous concerns. In particular they now present a more balanced view of the various drivers of phase behavior, including caveats for the role of magic numbers. The work is a very interesting contribution to the field of phase separation and hopefully will stimulate experimental testing of the models presented.